# Traumatic Events and Post-Traumatic Stress Disorder in Adolescents with Severe Eating Disorder Admitted to a Day Care Hospital

**DOI:** 10.3390/nu17132125

**Published:** 2025-06-26

**Authors:** Arturo Rodríguez-Rey, Flavia Piazza-Suprani, Elisabet Tasa-Vinyals, Maria Teresa Plana, Itziar Flamarique, Mireia Primé-Tous, Elena Moreno, Ines Hilker, Ester Pujal, Esteban Martínez, Susana Andrés-Perpiñá

**Affiliations:** 1Department of Child and Adolescent Psychiatry and Psychology, Institut Clinic de Neurociències, Hospital Clínic de Barcelona, 08036 Barcelona, Spaintasa@clinic.cat (E.T.-V.);; 2Fundació de Recerca Clínic Barcelona-Institut d’Investigacions Biomèdiques August Pi i Sunyer (IDIBAPS), 08036 Barcelona, Spain; 3Faculty of Medicine, University of Barcelona, 08036 Barcelona, Spain; 4Centro de Investigación Biomédica en Red de Salud Mental, CIBERSAM—ISCIII, 28029 Madrid, Spain; 5Faculty of Psychology, University of Barcelona, 08036 Barcelona, Spain

**Keywords:** eating disorders, adolescents, traumatic events, post-traumatic stress disorder

## Abstract

Background: It is common for patients with eating disorders (ED) to report traumatic experiences early in their lifetime. The objective of this study was to explore the presence and types of traumatic events and the comorbidity with Post-Traumatic Stress Disorder (PTSD) in a sample of adolescents with severe ED. Method: A total of 118 adolescents treated at our Eating Disorders Day Care Hospital (EDDCH) were systematically evaluated for depressive symptoms, disordered eating, early traumatic experiences, and presence of PTSD. Likewise, various clinical variables were collected including comorbidities, age upon ED diagnosis, number of hospital admissions, presence of non-suicidal self-injury, and suicide attempts. Results: Seventy-seven (65.3%) of adolescents of the total sample reported exposure to four or more traumatic events (bullying, psychological abuse, and sexual abuse being the most common). Fifty-seven of them (48.3% of the total sample) scored significantly high in PTSD assessment. Patients with ED and comorbid PTSD (PTSD+) presented higher disordered eating (***p* < 0.001**) and depressive symptoms (***p* < 0.001**) and also a higher prevalence of both non-suicidal self-injury (*p* = 0.031) and suicide attempts (*p* = 0.004). The depressive symptoms, measured with the CDI, emerged as an independent predictor of belonging to the PTSD+ group, explaining 22.9% of the variance. Conclusions: It is imperative to systematically screen adolescents with severe ED for traumatic events and PTSD, especially in patients presenting with more depressive symptoms and suicidal or non-suicidal self-injury behaviours, since this subset of patients could be at a higher risk of PTSD. Offering specific psychotherapeutic care targeting PTSD and/or posttraumatic symptoms in addition to the ED standard of care could arguably improve the prognosis of the ED in comorbid patients.

## 1. Introduction

Eating disorders (EDs) are estimated to have a lifetime prevalence of around 4–5% in Spanish young women and adolescents [1], similar to other developed countries [2,3], some of their symptoms being fairly prevalent in the general population [4]. EDs are severe mental disorders associated with high treatment costs, high comorbidity [5,6,7,8], and a very high mortality rate [9]. EDs are also associated with the occurrence of non-suicidal self-injury (NSSI) [10], increased risk of mortality from completed suicide [11,12], and suicide attempts [13].

It is currently rather common knowledge that EDs are not only a matter of irregular eating habits. It was almost four decades ago when studies began to relate exposure to traumatic experiences in early stages of life to a greater likelihood of developing an ED later on [14], which could be understood as a possible long-term effect of having suffered from abuse and other kinds of trauma in childhood [15,16]. Likewise, a strong comorbidity exists between Post-Traumatic Stress Disorder (PTSD) and EDs [17], similarly to what occurs in other psychiatric disorders [18]. After these initial approaches, in the past few years the relevance of having a traumatic childhood history has continued to be emphasised as a risk factor for developing an ED later in life [19,20]. Indeed, as biopsychosocial phenomena, EDs are rarely disconnected from the subject’s biographical background.

Overall, the most significant finding so far is that rates of ED are generally higher in people who have experienced early-life traumatic events and PTSD [21,22]. The prevalence of traumatic events in ED patients has been found to range from 37% to 100% [21]. Likewise, studies exploring the presence of PTSD in ED samples have found a prevalence range of 4% to 52% [23]. Several meta-analyses show a relationship between having suffered from childhood maltreatment and PTSD and having an ED [24,25,26]. Both meta-analyses [24,25] were carried out in order to specifically elucidate whether there was a relationship between child sexual abuse (CSA) and subsequently presenting an ED and showed a significant positive relationship between both experiences. The authors reported as a main limitation the heterogeneity of the studies, and postulated that CSA is not a specific risk factor for developing an ED. In addition, other studies that focus on other traumatic experiences, such as emotional and/or physical child neglect, show similar results [26,27], and emphasise the enormous relevance of assessing the possible existence of a childhood neglect history in ED patients, understood as an early traumatic event.

Hence, recent studies indicate that different types of trauma seem to be related to EDs, and that the majority of patients with ED report a history of interpersonal trauma. There are many types of trauma that can be associated with ED, including physical abuse and assault, sexual assault and harassment, emotional abuse, emotional and physical neglect, teasing and bullying [19]. Bullying may be a risk factor for developing and maintaining an ED, as has been previously determined [28] especially in adolescence, a period when both problems are most frequently encountered.

Endurance of traumatic events seems to be also related to greater severity and earlier onset of the ED [29]. There is evidence for an association between the number and types of trauma endured and the severity of ED and PTSD symptoms [19,30]. Therefore, knowing the presence and nature of the traumatic experience(s) that contributes to an ED could be critical to developing and implementing effective treatments. As mentioned above, in many cases the ED develops on the basis of untreated traumatic experiences from the past and consequently is particularly important to assess eventual traumatic events and post-traumatic stress symptoms and/or disorder (PTSD) in the process of treating an ED, particularly in severe cases [22].

The main purpose of this study was to explore the presence and types of traumatic events, as well as the comorbidity with PTSD, in a sample of adolescents with severe Anorexia Nervosa and Bulimia Nervosa admitted to an Eating Disorder Day Hospital (EDDCH). In light of the current evidence available, we expected to find a high prevalence of trauma and PTSD in our sample. Consequently, we hypothesised that comorbidity with PTSD would be associated with more severe ED symptomatology and greater overall clinical severity in our sample.

## 2. Methods

### 2.1. Participants

The study sample consisted of 118 children and adolescents aged between 11 and 17 years with a primary diagnosis of Anorexia Nervosa (AN) or Bulimia Nervosa (BN) according to DSM-5 criteria [31]. These diagnoses were obtained after a comprehensive clinical interview and examination of the patient’s physical and mental state by psychiatrists and clinical psychologists specialising in ED. The study participants were recruited consecutively from August 2020 to August 2022, from the Eating Disorder Day Care Hospital (EDDCH) at the Child and Adolescent Psychiatry and Psychology Department of the Hospital Clinic of Barcelona (Spain). The EDDCH is a therapeutic device specific for patients suffering from EDs of moderate or high severity. Some of the patients have previously attempted treatment at other settings in our public and universal healthcare system and/or in the private healthcare network, but unfortunately such previous therapeutic attempts have failed to improve the patient’s clinical condition. Criteria for a patient to be admitted to our EDDCH, and therefore in the study, are (1) present with a diagnosis of a severe AN or BN requiring intensive treatment; (2) not showing significant improvement after 6 months of outpatient treatment; (3) having a diagnosis of AN or BN that, despite not being severe in itself, presents with other severe psychiatric symptoms or comorbidity; (4) being underage (<18 years old); and (5) not present with severe intellectual disability. Severity criteria for AN and BN in children and adolescents, as in adults, goes far beyond weight loss. An important difference between underage and adult patients, however, is that weight loss and weight status are not based on rough Body Mass Index (BMI) values, but on BMI percentile variations. Nevertheless, severity is best assessed clinically biological (e.g., bradycardia, dyselectrolytemia, hematemesis…) and/or psychopathological (e.g., suicidal ideation, psychotic symptoms…) examination.

The hospital ethical committee approved this retrospective study (Approval Code: HCB/2022/1118. Approval Date: 20 January 2023).

### 2.2. Clinical and Sociodemographic Variables

The clinical and sociodemographic variables were extracted from the hospital’s clinical management software. The variables collected were age, diagnoses, length of stay in EDDCH (until August 2022), number of inpatient unit admissions and length of inpatient stay until August 2022, history of suicide attempts (yes/no), history of non-suicide self-injury (yes/no), and onset age of the ED.

### 2.3. Measures

The evaluation was conducted in two phases:

Phase 1. The following questionnaires were administered to all patients admitted to the EDDCH upon collection of informed consent:Child Depression Inventory, adapted to Spanish (CDI) [32,33]. The CDI is a 27-item self-report questionnaire which assesses affective, cognitive, and behavioural symptoms of depression. Each item consists of three statements graded in order to increase severity from 0 to 2. The cut-off point is 19. It has shown relatively high levels of internal consistency, retest reliability, and convergent validity.Eating Attitude Test, adapted to Spanish (EAT-40) [34,35]. The EAT-40 is a 40-item self-administered questionnaire with six possible answers ranging from ‘never’ to ‘always’. The total test score distinguishes between anorexic patients and the normal population, and between bulimics and the normal population. The Spanish version [34] has shown to achieve adequate sensitivity at a cut-off point of 20. It has also demonstrated good internal consistency.Early Trauma Inventory Self-Report Short Form (ETI-SR-SF) [36,37]. The ETI-SR-SF systematically assesses exposure to general trauma (11 items), physical abuse (5 items), emotional abuse (5 items), and sexual abuse (6 items). It has shown good internal consistency and psychometric properties in the Spanish adult population. The questionnaire offers two response options, one for minors and one for those over 18 years old. In our case, we only used the former, since our total sample is aged under 18. As the ETI-SR-SF is not validated in Spanish for the child and adolescent population, we used a cut-off point of 4 to evaluate the possible presence of PTSD following the validation in Spanish for an adult female population [36]. In doing to, we rely on previous findings that demonstrate that having endured at least four adverse childhood experiences (ACEs) is an important risk factor for many health conditions across the life span [38].

Phase 2. Patients were assessed for the possible presence of PTSD. This part of the evaluation was carried out as follows: Individual interview with a clinician to reach an agreement with the patient on which of the events marked in the ETI-SR-SF had been the most traumatic for them.Post-Traumatic Stress Disorder Symptom Severity Scale-Revised (EGSR) [39]. The EGSR is a 21-item structured interview based on DSM-5 criteria and is used to assess the severity of PTSD symptoms (re-experiencing; avoidance; cognitive and mood disturbances; hyperarousal). Patients should complete this questionnaire considering PTSD symptoms related to the traumatic event selected after the Phase 2 interview conducted with a clinician.

### 2.4. Data Analysis

Descriptive summary statistics (means, standard deviations, frequencies, percentages) were used to describe the sociodemographic and clinical characteristics of the sample. For the comparison study, patients were divided into two groups: patients with ED and PTSD (PTSD+) and patients with ED without PTSD (PTSD−). The Kolmogorov–Smirnov and Levene tests were applied to assess the normality of the sample distribution and the equality of variances. Student’s t test or Mann–Whitney U tests were used to compare the means of continuous variables between the two groups of patients (PTSD+ and PTSD−) depending on parametric assumption fulfilment. Categorical clinical variables were analysed using the Chi-square test. To study the predictive capacity of some clinical variables on the diagnosis of PTSD, a logistic regression model was fitted using a forward stepwise method. Variables that showed significant differences between PTSD+ and PTSD− groups were included as potential predictors. Terms such as ‘predictor variable’ and ‘explained variance’ were used in a statistical sense, without implying causality. A *p*-value of ≤0.05 was the significance level established for all analyses. All statistical analyses were performed using SPSS 19.0 for Windows.

## 3. Results

### 3.1. Participant’s Characteristics

The total sample was of 118 patients, of which 98.3% were female. The mean age was 14.9 years old (SD = 1.53) within the range of 11 and 17 years old. In terms of ethnic origin, 87.3% of the patients were of Mediterranean ascent, and 9.3% from South American ascent.

Most of the patients presented a DSM-5 diagnosis of Anorexia Nervosa restrictive type (AN-R) (n = 88, 74.6%) or Anorexia Nervosa purgative type (AN-P) (n = 15, 12.7%). The remaining patients had a diagnosis of Bulimia Nervosa (BN) (n = 7, 5.9%) or Eating Disorders Not Otherwise Specified (EDNOS), mostly atypical anorexia (n = 8, 6.8%). Additionally, a relevant number of patients (n = 86, 72.8%) had comorbid mental health diagnoses: 55.9% (n = 66) of the patients had one comorbid diagnosis and 16.9% (n = 20) had two comorbid diagnoses. The most frequent comorbid diagnoses were depressive disorder, anxiety disorder, and Obsessive–Compulsive Disorder (OCD) (Table 1).

### 3.2. Phase 1: Assessment of Depressive Symptoms, ED Symptoms, and Traumatic Events

The first aim of this study was to detect the presence and number of lifetime traumatic experiences in the total sample. The number of traumatic events reported by the patients showed a mean of 5.69 events (range 0–19). Seventy-seven patients (65.3% of the total sample) scored 4 or higher in the ETI-SR-SF. The results of the battery of tests administered in Phase 1 of the evaluation to all patients showed high scores in the CDI (mean = 27.5 (SD = 8.7)) and in the EAT (mean = 65.8 (SD = 22.6)).

### 3.3. Phase 2: Assessment of PTSD

The second aim was to confirm how many patients met PTSD criteria and which was the most severe or shocking traumatic event for each individual evaluated. For this purpose, the most significant traumatic event was selected in an individual interview between each patient and a clinician, based on the patient’s previous responses to the ETI-SR-SF. The patient selected which of the events indicated in the questionnaire had had the greatest impact in their life, according to their subjective perception. Fifty-seven (48.3% of the total sample) showed scores above the cut-off point of EGSR, indicating possible presence of PTSD. The mean score of the sample for the EGSR was 31.9 (SD = 13.8).

### 3.4. Type of Traumatic Events Reported

The most relevant traumatic events were identified in the analysis of the answers in an individual interview with the patients. Out of the 57 patients who met PTSD criteria, 16 identified only one type of traumatic event as significantly impacting their life, 30 identified two types of traumatic events, and 11 indicated three types of traumatic events. The most frequent traumatic events in this sample of patients were bullying, psychological abuse or neglect, and sexual abuse (Table 2).

### 3.5. Relationship Between PTSD and Severity of Disorder

The third aim of this study was to compare the clinical characteristics of adolescents with ED and comorbid PTSD (PTSD+) and patients with ED without comorbid PTSD (PTSD−). As shown in Table 3, no significant differences were found between the two groups in quantitative variables such as age, number of comorbid diagnoses, number of admissions to the EDDCH or inpatient unit, or number of days spent in the EDDCH or the inpatient unit. However, significant differences were found between both groups in the scores of the EAT-40 and CDI questionnaires, showing more presence of ED and depressive symptoms in the PTSD+ group.

Regarding the qualitative variables, contingency tables were computed for the dichotomous variables suicide attempts and non-suicidal self-injury (NSSI). Both variables showed significant results, indicating that patients with PTSD+ presented higher percentages of NSSI and suicide attempts (Table 4).

A logistic regression was calculated with significant variables between groups PTSD− and PTSD+. The CDI and EAT scores and the presence of NSSI and suicide attempts were introduced in the model, but only the CDI score emerged as an independent predictor of belonging to the group PTSD+, explaining 22.9% of the variance (adjusted R^2^ = 0.229, Exp(B) = 1.125, ***p* < 0.001**).

## 4. Discussion

This study shows that almost half of the adolescents with severe ED treated in a specialised high-intensity facility have undiagnosed PTSD. These subgroups of comorbid patients experience more depressive symptoms, as well as more suicidal attempts and NSSI.

In our sample, the frequency of politraumatisation (defined as exposure to four or more lifetime traumatic events) was a relevant 65%, which is similar to figures reported in previous studies [21,40,41]. This finding confirms that a significant number of adolescents with ED have endured a significant number of traumatic experiences, at least in high-intensity therapeutic settings. Regarding the types of traumatic events, when we compare interpersonal and non-interpersonal trauma, our results are consistent with previous studies in that interpersonal trauma is most frequently reported [23]. We believe that this is relevant, since interpersonal traumas are known to produce more severe post-traumatic symptoms [30]. Unlike other studies, ours did not find sexual assault to be the most prevalent trauma type in our sample, although it occurred in not less than 28.1% of the patients with PTSD. Among the different types of traumatic events that we evaluated, the most frequent was bullying, with 47.3% among those patients with PTSD, followed by psychological abuse or neglect, with 29.8%. Therefore, because of the high prevalence of bullying compared to other types of traumatic experiences, bullying victimisation may be a specific risk factor for developing and maintaining an ED [28] in the adolescent population [42]. This appears coherent to us, as bullying meets many of the known criteria for an event to be highly traumagenic: it is an interpersonal experience, usually inflicted peer-to-peer during neurodevelopmental age, often overlooked and long-lasting (despite relatively recent antibullying campaigns in ours and other countries). Moreover, we might argue that bullying can be conceptualised and experienced as a unique, intimate-yet-exposed form of victimisation, opposite to many other sorts of violence that Western children/adolescents are familiar with present-day. This could lead bullying to impact children and adolescents in ways qualitatively different from other forms of trauma considered in this study, such as parental abuse or neglect, which tends to happen behind closed doors and be inflicted from an adult who holds a position of power to a child/adolescent who does not.

Regarding the diagnosis of PTSD, 48% of the total sample met criteria for the diagnosis of comorbid PTSD, similarly to what has been reported in other studies [43,44]. Interestingly, none of the patients with PTSD had been diagnosed before this assessment, which is why they were only receiving standard-of-care for ED treatment.

Our results point out that adolescents with severe ED and comorbid PTSD show more depressive and ED symptomatology, as well as more suicide attempts and NSSI, similarly to data reported in other studies [45,46]. However, we found no significant correlation with other variables such as age, number of comorbid diagnoses, number of days admitted in EDDCH and/or in general hospitalisation, number of admissions in EDDCH, or number of days hospitalised. This could be explained due to the relative homogeneity of our sample in terms of age, but also the fact that the majority of the patients already had at least one documented comorbidity with another mental disorder, and the fact that the sample was exclusively made up of severe ED patients, who in general are characterised by longer recovery times.

Adolescents and young adults, particularly females, with EDs are highly vulnerable to the development of anxiety and depressive disorders. Moreover, comorbid anxiety or depression in ED patients is associated with greater symptom severity, poorer prognosis, and a greater burden of illness [47]. In the present study, a higher score in depression was a predictor variable of comorbid PTSD. This finding could indicate that, although it is important to evaluate the presence of PTSD in all patients with ED, specific assessment might be especially relevant in those showing increasingly more severe depressive symptoms.

From a clinical perspective it must be underlined that, despite advances achieved in recent decades, the therapeutic response to different approaches for EDs remains far from being optimal [48,49,50], and in AN treatment response is arguably rather weak and inconsistent [51]. This poor response to conventional ED treatment could be modulated by a high prevalence of early traumatic events and PTSD in severe ED patients. Various studies [41,52,53] have shown that ED patients with past traumatic experiences show poorer therapy outcomes and are more likely to drop out of treatment and relapse. As various authors have recently proposed [54], individualising treatments based on the patients’ different clinical characteristics seems the best approach in terms of improving future management for EDs. Properly identifying PTSD in a patient may thus provide the opportunity to offer the gold standard treatment for this disorder (e.g., Eye Movement Desensitisation and Reprocessing—EMDR, or Trauma-Focused Cognitive Behavioural Therapy—TF-CBT) and therefore help achieve an optimal, tailored and more effective treatment in a significant proportion of cases. Of course, just as any other exploration or intervention, routine trauma assessment shall be carried out by trained specialist clinicians in appropriate timing and setting, ensuring the patient’s safety and clinical stability.

## 5. Conclusions

In conclusion, the results obtained in this study confirm that a high percentage of adolescents with severe ED have experienced four or more lifetime traumatic events, and that comorbidity with PTSD is highly prevalent. Likewise, the types of traumatic events experienced by ED patients are mostly interpersonal traumas, with bullying being the most prevalent. The results show that suffering from PTSD is related to more severe eating and depressive symptoms and higher percentages of NSSI and suicide attempts in patients with ED. On the other hand, our study highlights that PTSD in ED patients is arguably underdiagnosed even in more intensive treatment settings such as an EDDCH. These findings suggest that the presence of traumatic events and PTSD should be actively investigated in ED patients as a requirement in order to provide suitable, comprehensive and tailored interventions that specifically address trauma when needed, particularly in patients presenting with more and/or more severe depressive symptomatology. Our study also provides a possible explanation of the suboptimal therapeutic results often observed in some patients undergoing ED therapeutic care alone, as PTSD symptomatology may worsen ED symptomatology and make it more complex and more resistant to traditional therapeutic approaches. Lastly, our findings could prove beneficial to prevent retraumatisation or the development of further psychopathologies in at-risk, vulnerable adolescents by trauma-unaware clinicians and/or therapeutic settings.

## 6. Limitations

This study is, of course, not free from methodological limitations. First, the sample was collected from a single treatment centre and is primarily of Mediterranean ascent, which calls for careful interpretation and generalisation of the results given the risk of selection bias. In this regard, however, it must be noted that Spain has a public, universal healthcare system that might arguably reduce selection bias related to socioeconomical issues. Second, the cross-sectional design prevents us from establishing any causal relationships between traumatic events and ED severity or PTSD comorbidity. Additionally, readers should also bear in mind that the validity of the ETI-SR-SF has not been fully tested in adolescents, which may affect the accuracy of the traumatic event assessment carried out in this study, despite our best efforts to apply available data based on adult female population in a thoughtful and coherent manner. Future research would do well in incorporating objective, physiological data (e.g., biological markers of ED severity) to the study design, and further explore cultural issues that might be related to trauma prevalence.

## Figures and Tables

**Table 1 nutrients-17-02125-t001:** Comorbid diagnoses in the sample.

Comorbid Diagnoses	No	%
Depression	34	28.8
Anxiety disorder	16	13.6
Obsessive compulsive disorder	15	12.7
Borderline personality disorder	4	3.4
Oppositional defiant disorder	4	3.4
Substance use disorder	3	2.5
Social phobia	3	2.5
Total	86	72.8

**Table 2 nutrients-17-02125-t002:** Most relevant types of traumatic events identified in the patients with PTSD+ through the Phase 2 interview (n = 57).

Traumatic Events	n	%
Bullying	27	47.3
Psychological abuse or neglect	17	29.8
Sexual abuse	16	28.1
Sickness in the family	14	24.6
Physical abuse	12	21.1
Family violence witness	10	17.5
Accidents or natural catastrophes	2	3.5
Own sickness or hospitalisation	1	1.8

**Table 3 nutrients-17-02125-t003:** Comparison of quantitative variables between patients with ED and PTSD comorbid (PTSD+) and patients with ED but without PTSD (PTSD−).

Variables	PTSD+ (n = 57) Mean (SD)	PTSD− (n = 61) Mean (SD)	U Mann–Whitney	*p*
Age	15 (1.5)	14.8 (1.6)	1583.0	0.391
Nº of comorbid diagnoses	0.8 (0.7)	0.7 (0.8)	1607.5	0.444
Days admitted in EDDCH	166.1 (78.1)	176.1 (153.4)	1441.5	0.320
Nº of admissions in EDDCH	1.6 (0.8)	1.6 (1.2)	1553	0.249
Nº of hospitalisation	1.4 (1.2)	1.3 (1.3)	1586	0.390
Nº of days hospitalised	51.7 (71.8)	44.5 (51.9)	1680.5	0.753
EAT-40 total	73.6 (19.1)	58.7 (23.3)	Students-*t* test 974.5	** *p* ** **<0.001**
CDI	31.2 (8.4)	24.1 (7.5)	−4.8	<0.001

**Table 4 nutrients-17-02125-t004:** Comparison of qualitative variables between patients with ED and PTSD comorbid (PTSD+) and patients with ED but without PTSD (PTSD−).

	PTSD+ (n = 57)	PTSD− (n = 61)	Chi-Square	*p*
Suicide attempts	21 (36%)	9 (14.7%)	8.077	0.004
Non-suicidal self-injury	39 (67.2%)	31 (50.8%)	4.632	0.031

## Data Availability

Authors agree to make data and materials supporting the results or analyses presented in their paper available upon reasonable request due to logistic and ethical reasons.

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
