# Peer review of "Traumatic Events and Post-Traumatic Stress Disorder in Adolescents with Severe Eating Disorder Admitted to a Day Care Hospital"

_nutrients, 2025, doi:10.3390/nu17132125_

Round 1

Reviewer 1 Report

Comments and Suggestions for Authors

Current paper deals with an interesting topic which has been largely explored by previous literature in the adult ED population. The present paper has the merit of addressing adolescent population and severe cases of EDs and thus it adds some new specific information. 

Moreover a revision of some contents is necessary prior to the publication of the manuscript.

Title: to evidence the novelty of the findings the title should underline that the paper refers to "severe" ED adolescents.

Abstract: in the results section also the overall of all traumatic experiences should be given, not only those who are more than 4.

Introduction: since the intro does not specify what EDs are included (since 2013 the DSM includes also the BED and ARFID diagnoses that are very relevant for adolescence) it the authors intend to treat the topic of abuses for ALL Eds they should alòso cite some recent research on BED psychological and physical abuse, instead they should specify that only AN and BN are included in the research.

Methods: the authors should clearly indicate what are the DSM diagnoses included in the paper. Moreover they should also indicate the DSM degree of severity among inclusion criteria for each diagnosis or an alternative instrument to define the "severity" of the included adolescents. The authors use the ETI-SR-SF instrument and admit that it is not validated for their population: the choice of choosing a screening cutoff of 4 should be better explained with respect to the adult literature. Moreover the authors should underline this limit in the specific section. Table 1 reports the prevalence of comorbid diagnosis, nevertheless only depression is assessed with a specific tool, the authors should better specify how the other diagnoses are provided. Table 2 reports a series of traumatic events without specification of the assessment method, was it the ETI-SR-F? 

All tables need the same format (possibly without lines in the body of the table)

Discussion: it is quite sketchy, and it lacks a limits section. More comparison with the current literature should be made and some more discussion of the only relevant difference (depression) between the analyzed groups should better valorize the findings. The discussion of clinical implications should be more specific and centered to the research finding.

Comments on the Quality of English Language

The paper needs linguistic revision because of morphological and syntactical errors (e.g. abstract lines 23 'traumatic events early' and 30 'group more vulnerable', but also intro line 42 'are rarely only related to', etc.). 

Author Response

For research article

Response to Reviewer 1 Comments

1. Summary

2. Questions for General Evaluation

Reviewer’s Evaluation

Response and Revisions

Does the introduction provide sufficient background and include all relevant references?

Can be improved

Introduction has been reviewed and improved

Are all the cited references relevant to the research?

Yes

Is the research design appropriate?

Yes

Are the methods adequately described?

Can be improved

Methods have been explained in greater detail

Are the results clearly presented?

Yes

Are the conclusions supported by the results?

Yes

3. Point-by-point response to Comments and Suggestions for Authors

Comment 1: Title: to evidence the novelty of the findings the title should underline that the paper refers to "severe" ED adolescents.

Response 1: Agree. The title has been modified as suggested. The new title is: “Traumatic Events and Post-Traumatic Stress Disorder in Adolescents with Severe Eating Disorder Admitted to a Day Care Hospital”. We have also decided to use the full expression Day Care Hospital in order to improve the paper linguistically, as suggested by both reviewers.

Comment 2: Abstract: in the results section also the overall of all traumatic experiences should be given, not only those who are more than 4.

Response 2: We have reviewed this point. Thanks!

Comment 3: Introduction: since the intro does not specify what EDs are included (since 2013 the DSM includes also the BED and ARFID diagnoses that are very relevant for adolescence) it the authors intend to treat the topic of abuses for ALL EDs they should also cite some recent research on BED psychological and physical abuse, instead they should specify that only AN and BN are included in the research.

Response 3: In our EDDCH most of our patients are diagnosed with Anorexia Nervosa or Bulimia Nervosa. Therefore, these two diagnoses are the ones in our study. We have better explain this point in the introduction and method sections. Thank you for the suggestion.

Comment 4: They should also indicate the DSM degree of severity among inclusion criteria for each diagnosis or an alternative instrument to define the "severity" of the adolescents included.

Response 4: The diagnosis of ED in our unit is reached after a comprehensive clinical interview conducted by a psychiatrist and/or a clinical psychologist specializing in ED. The professional goes through an extensive anamnesis with the underage patient’s family and also obtains anamnesis and physical and psychopathological exam of the patient. Our unit specializes in moderate or high severity ED cases. By the time patients are remitted to EDDCH, treatment has already been attempted in lower-severity facilities within our public universal health system, such as community health centres or general child and adolescent mental health units, having shown a poor therapeutic response. We explain in more detail the inclusion criteria of our day hospital, in Method.

Comment 5: The authors use the ETI-SR-SF instrument and admit that it is not validated for their population: the choice of choosing a screening cutoff of 4 should be better explained with respect to adult literature. Moreover, the authors should underline this limit in the specific section.

Response 5: We explain this point in more detail. A Limits section has been added to the manuscript.

Comment 6: Table 1 reports the prevalence of comorbid diagnosis, nevertheless only depression is assessed with a specific tool, the authors should better specify how the other diagnoses are provided.

Response 6: The diagnosis of ED and comorbid diagnoses are made through clinical interviews by psychiatrists and clinical psychologists.

Comment 7: Table 2 reports a series of traumatic events without specification of the assessment method, was it the ETI-SR-F? 

Response 7: The most significant traumatic events are selected in an individual interview with each patient, based on their responses on the ETI-SR-SF. The patient selects which of the events indicated in the questionnaire has had the greatest impact in their life, from their point of view. We have revised and improved the description of Phase 2 method.

Comment 8: All tables need the same format (possibly without lines in the body of the table)

Response 8: Tables have been reformatted.

Comment 9: Discussion: it is quite sketchy, and it lacks a limits section. More comparison with the current literature should be made and some more discussion of the only relevant difference (depression) between the analysed groups should better valorise the findings. The discussion of clinical implications should be more specific and centred to the research finding.

Response 9: The discussion has been extended and more linked to previous findings. We've also added a limitations section and expanded the clinical implications of our work. In our view, the quality of this section has significantly improved. Thanks for pointing out and helping us achieve this improvement.

Reviewer 2 Report

Comments and Suggestions for Authors

I read the author's submitted article with great interest. This study explores the comorbidity of traumatic events and post-traumatic stress disorder (PTSD) in adolescent patients with eating disorders (ED). The research design is reasonable, the data are detailed, and the study holds certain clinical significance. Below are my evaluations and suggestions for the article:

Significance and Innovation: This study focuses on the comorbidity of traumatic events and PTSD in adolescent ED patients, addressing a gap in the field. The results indicate that nearly half of severe ED patients have PTSD, and traumatic events (particularly bullying and psychological abuse) are significantly correlated with the severity of ED symptoms. This finding provides important evidence for the comprehensive assessment and individualized treatment of ED patients in clinical practice. Additionally, the study reveals the independent predictive role of depressive symptoms in PTSD comorbidity, offering direction for future intervention research.

Methods and Design: The study adopts a cross-sectional design, with a moderate sample size (n=118) drawn from a day treatment center at a hospital in Spain. The sample is predominantly female (98.3%), consistent with the gender distribution of ED. Standardized questionnaires (e.g., CDI, EAT-40, ETI-SR-SF) and clinical interviews (EGSR) were used to assess depressive symptoms, eating disorder symptoms, traumatic events, and PTSD, demonstrating methodological rigor. However, the study does not clearly specify the randomness of sample selection, which may introduce selection bias. Furthermore, all assessments were self-reported or interview-based; future research could incorporate physiological indicators or multidimensional assessment tools to enhance objectivity. In lines 150-152, I suggest adding a reference to previous literature focusing on mental health [doi: 10.3390/nu16060777] to support the statement: "To study the predictive capacity of some clinical variables on the diagnosis of PTSD, a logistic regression model was fitted using a forward stepwise method."

Results and Discussion: The findings support the authors' hypothesis that PTSD comorbidity is associated with more severe ED symptoms, depressive symptoms, and higher rates of non-suicidal self-injury and suicide attempts. These results align with previous research, further confirming the complex relationship between trauma and ED. However, the study did not find a correlation between PTSD comorbidity and clinical variables such as hospitalization duration or admission frequency, which may be due to the high homogeneity of the sample. Future studies could expand the sample range to validate this finding. The discussion section provides a thorough interpretation of the results but does not delve deeply into the potential mechanisms of bullying as a primary traumatic event. It is recommended to supplement this with relevant theories or literature.

Limitations: The study has the following limitations: First, the sample comes from a single treatment center and is primarily of Mediterranean descent, which may limit the generalizability of the results. Second, the cross-sectional design cannot establish causal relationships between traumatic events and ED or PTSD. Additionally, the validity of the ETI-SR-SF in adolescents has not been fully verified, which may affect the accuracy of traumatic event assessment. While the authors mention some limitations in the discussion, the impact of cultural factors on the results is not sufficiently addressed. It is recommended to expand on this in the revision.

Writing and Structure: The article is well-structured and logically coherent, though some paragraphs are slightly verbose and could be further streamlined. The introduction provides a comprehensive review of the literature on the relationship between ED and trauma but could include more discussion on the unique aspects of the adolescent population. The methods section should include additional details on ethical approval and informed consent. The results section presents data clearly, but the table formats need to be standardized (e.g., decimal places for percentages). The references are sufficient in number, but some citation formats require proofreading (e.g., the abbreviated journal name in Reference 3 is inconsistent).

Author Response

For research article

Response to Reviewer 2 Comments

1. Summary

2. Questions for General Evaluation

Reviewer’s Evaluation

Response and Revisions

Does the introduction provide sufficient background and include all relevant references?

Yes

Are all the cited references relevant to the research?

Yes

Is the research design appropriate?

Can be improved

Design have been explained in more detail.

Are the methods adequately described?

Can be improved

Methods have been described in more detail.

Are the results clearly presented?

Yes

Are the conclusions supported by the results?

Yes

3. Point-by-point response to Comments and Suggestions for Authors

Comment 1: Methods and Design: The study adopts a cross-sectional design, with a moderate sample size (n=118) drawn from a day treatment center at a hospital in Spain. The sample is predominantly female (98.3%), consistent with the gender distribution of ED. Standardized questionnaires (e.g., CDI, EAT-40, ETI-SR-SF) and clinical interviews (EGSR) were used to assess depressive symptoms, eating disorder symptoms, traumatic events, and PTSD, demonstrating methodological rigor. However, the study does not clearly specify the randomness of sample selection, which may introduce selection bias.   

Response 1: The sample for this study consisted of patients who attended our EDDCH for treatment for their ED. The participants were recruited consecutively from August 2020 to August 2022. Our unit specializes in moderate or high severity ED cases. By the time patients are remitted to EDDCH, treatment has already been attempted in lower-severity facilities within our public universal health system, such as community health centers or general child and adolescent mental health units, having shown a poor therapeutic response. We explain in more detail the inclusion criteria of our day hospital, in Method. The hospital's ethical committee approved this study. The sample of our study is representative of the population ascribed to our EDDCH, and representative of adolescents with a severe form of AN or BN. This point is better explained in Methods section. 

Comment 2: Furthermore, all assessments were self-reported or interview-based; future research could incorporate physiological indicators or multidimensional assessment tools to enhance objectivity.

Response 2: Thank you for this very valuable suggestion for future research projects. We will definitely bear it in mind.

Comment 3: In lines 150-152, I suggest adding a reference to previous literature focusing on mental health [doi: 10.3390/nu16060777] to support the statement: "To study the predictive capacity of some clinical variables on the diagnosis of PTSD, a logistic regression model was fitted using a forward stepwise method."

Response 3 Thank you for the suggestion. We have carefully reviewed the reference you suggest and unfortunately fail to see how it might link in a significant way to the point of our study. It is true that both studies share certain methods, but those are rather standard (regression) and the variables in both studies are conceptually apart. Therefore, after careful consideration, we have decided not to include it since we do not see a clear benefit for our article’s audience in doing so.

Comment 4: Results and Discussion: The findings support the authors' hypothesis that PTSD comorbidity is associated with more severe ED symptoms, depressive symptoms, and higher rates of non-suicidal self-injury and suicide attempts. These results align with previous research, further confirming the complex relationship between trauma and ED. However, the study did not find a correlation between PTSD comorbidity and clinical variables such as hospitalization duration or admission frequency, which may be due to the high homogeneity of the sample. Future studies could expand the sample range to validate this finding.

Response 4: As we explain in Comment 1, patients are recruited consecutively from our day hospital. In our setting, patients are rather homogenous regarding diagnosis, since virtually all patients meet diagnostic criteria for AN or BN, moderate to severe. The results of this study are generalizable to patients with similar characteristics or devices with similar severity.

Comment 5: The discussion section provides a thorough interpretation of the results but does not delve deeply into the potential mechanisms of bullying as a primary traumatic event. It is recommended to supplement this with relevant theories or literature.

Response 5: The discussion has been rewritten around this very important point. We hope its quality has now improved.

Comment 6: The study has the following limitations: First, the sample comes from a single treatment center and is primarily of Mediterranean descent, which may limit the generalizability of the results. Second, the cross-sectional design cannot establish causal relationships between traumatic events and ED or PTSD. Additionally, the validity of the ETI-SR-SF in adolescents has not been fully verified, which may affect the accuracy of traumatic event assessment. While the authors mention some limitations in the discussion, the impact of cultural factors on the results is not sufficiently addressed. It is recommended to expand on this in the revision.

Response 6: A Limitations section has been added to the discussion section. Thank you for the valuable suggestion, since it is really useful for the reader to have all the limitations listed together.

Comment 7: The article is well-structured and logically coherent, though some paragraphs are slightly verbose and could be further streamlined. The introduction provides a comprehensive review of the literature on the relationship between ED and trauma but could include more discussion on the unique aspects of the adolescent population.

Response 7: Thank you for your insight. The introduction has been reviewed, and we have included more particular aspects of adolescence throughout the rest of the manuscript as well.

Comment 8: The methods section should include additional details on ethical approval and informed consent.

Response 8: Information about data collection and ethical approval is explained in more detail in the Methods section.

Comment 9: The results section presents data clearly, but the table formats need to be standardized (e.g., decimal places for percentages)

Response 9: Tables have been reformatted for coherence and uniformity.

Comment 10:  The references are sufficient in number, but some citation formats require proofreading (e.g., the abbreviated journal name in Reference 3 is inconsistent).

Response 10: References have been improved to the best of our ability.

4. Response to Comments on the Quality of English Language

Point 1: The English is fine and does not require any improvement.

Response 1: Thanks! However, the current version of the manuscript has been reviewed again by another English-speaking expert in a final attempt to improve language flow and grasp minor eventual mistakes.
